# Economic Aspects of Mechanical Pre-Treatment's Role in Precious Metals Recovery from Electronic Waste

Ervins Blumbergs [1,2,3], Andrei Shishkin [1,4,*], Karlis Markus [1], Vera Serga [5], Dmitri Goljandin [6], Artur Klauson [6], Vitalijs Abramovskis [4], Janis Baronins [4], Aleksej Zarkov [7,8] and Vladimir Pankratov [8]

1   ZTF Aerkom SIA, 32 Miera Str., LV-2169 Salaspils, Latvia; ervinblumbergs@gmail.com (E.B.); karlis@jic.lv (K.M.)
2   Institute of Physics, University of Latvia, 32 Miera Str., LV-2169 Salaspils, Latvia
3   Faculty of Civil Engineering, Riga Technical University, 21/1 Azenes Str., LV-1048 Riga, Latvia
4   Laboratory of Ecological Solutions and Sustainable Development of Materials, Institute of General Chemical Engineering, Faculty of Materials Science and Applied Chemistry at Riga Technical University, 3/7 P. Valdena, Str., LV-1048 Riga, Latvia; vitalijs.abramovskis@rtu.lv (V.A.); janis.baronins@rtu.lv (J.B.)
5   Institute of Materials and Surface Engineering, Faculty of Materials Science and Applied Chemistry, Riga Technical University, 3/7 P. Valdena Street, LV-1048 Riga, Latvia; vera.serga@rtu.lv
6   Department of Mechanical and Industrial Engineering, Tallinn University of Technology, Ehitajate Tee 5, 19086 Tallinn, Estonia; dmitri.goljandin@taltech.ee (D.G.); artur.klauson@taltech.ee (A.K.)
7   Institute of Chemistry, Vilnius University, 24 Naugarduko Street, LT-03225 Vilnius, Lithuania; aleksej.zarkov@chf.vu.lt
8   Institute of Solid State Physics, University of Latvia, 8 Kengaraga Str., LV-1063 Riga, Latvia; vladimirs.pankratovs@cfi.lu.lv
*   Correspondence: powdel.al.b@gmail.com or andrejs.siskins@rtu.lv

**Correction Statement:** This article has been republished with a minor change. The change does not affect the scientific content of the article and further details are available within the backmatter of the website version of this article.

**Abstract:** Printed circuit boards (PCBs) make up 3 to 5% of all electronic waste. The metal content of spent PCBs can reach 40%. They usually contain valuable metals, such as Ag, Au, and Pd, as well as other metals such as Cu, Sn, Pb, Cd, Cr, Zn, Ni, and Mn. However, the metallic part of a whole PCB is 40–60% including the Cu layers between the fiberglass–polymer layers. The paper describes the economics of the valuable metal (Ag, Au, Pd)-containing concentrate preparation from a raw PCB. We considered the influence of the pre-treatment method of PCBs before the extraction of valuable metals on the extraction self-cost change. The disintegration method is based on the high-energy impact of the particles of the material to be ground, thus causing the separation of the metallic components of the PCB. In the course of the work, single and double direct grinding using the method of disintegration was studied. For the calculation, the test batch of 10,000 kg of two types of PCB was taken for estimation of the self costs and potential profit in the case of complete valuable metals (Ag, Au, Pd) plus Cu extraction. It was shown that from 10,000 kg of studied PCB, it is possible to obtain 1144 and 1644 kg of metal-rich concentrate, which should be further subjected to electro-hydrochlorination for metals leaching. The novelty of this research lies in the fact that a technical and economic analysis has been carried out on a newly developed combined technology for processing electronic waste. This included mechanical processing and electrochemical leaching with the help of the active chlorine that is formed in situ. The real (not specially selected or prepared) waste PCBs were used for the process's economical efficiency evaluation. The main findings showed that despite the high content of Cu in the studied PCBs, the commercial value was insignificant in relation to the total income from the Ag, Au, and Pd sale. A correlation was established between the self-cost decrease after separative disintegration of PCBs by metal content increase (by specific metals such as Au, Ag, Pd, and Cu) with the metal potential yield after extraction.

**Keywords:** waste printed circuit boards; PCB; recycling; disintegrator milling; separation; precious metals; recovery; self-costs

## 1. Introduction

In 2023, it is estimated that the worldwide amount of waste electronic and electrical equipment (WEEE) will reach 57.4 million metric tons [1]. Of this, waste printed circuit boards (PCBs) represent the most economically attractive portion and account for approximately 3% of the total e-waste [2]. The Global E-waste Monitor 2020 reported that in 2019, approximately 53.6 million metric tons of WEEE was generated, showing a 21% increase over a five-year period since 2014. If this trend continues, it is predicted that e-waste will reach 74 million metric tons by 2030. This indicates an annual growth of 2 million metric tons, or approximately 3 to 4% annually. The problem is largely attributed to higher rates of electronic consumption, which are increasing by approximately 3% each year, along with shorter product lifecycles and limited repair options [3].

In Europe, where e-waste is extensively studied, it is estimated that an average household has 11 out of 72 electronic items that are either no longer in use or broken. Additionally, each European citizen hoards another 4 to 5 kg of unused electrical and electronic products annually before eventually discarding them. These figures highlight the scale of the e-waste problem within Europe and the need for the proper management and disposal of electronic devices [4]. In our rapidly advancing world of science and technology, the proliferation of electronic products has increased significantly. However, this trend has also contributed to the accelerated obsolescence of these products, resulting in a substantial amount of electronic waste (e-waste) [5]. Currently, e-waste production worldwide stands at 40–50 million tons per year, with an annual increase of 5 million tons [6]. Waste printed circuit boards (WPCBs), which constitute 4–7% of the total e-waste mass, play a critical role in electronic products [7]. Unfortunately, WPCBs are also the most complex and hazardous components of e-waste, containing toxic heavy metals (such as Pb, Cr, Cd, and Hg) and toxic organic substances (such as brominated flame retardants and polycyclic aromatic hydrocarbons) [8,9]. The accumulation and persistence of these substances in the environment pose a significant threat to both ecosystems and human health [10,11]. Without proper treatment, these pollutants can cause severe environmental damage.

However, WPCBs also present an opportunity for resource recovery, as they contain valuable metals such as copper (Cu), tin (Sn), iron (Fe), nickel (Ni), and zinc (Zn), as well as precious metals such as gold (Au) and silver (Ag). The metal content in WPCBs, particularly precious metals, exceeds that of primary mineral resources, making their recycling economically, socially, and environmentally beneficial [12]. The complex composition and structure of WPCBs make their recycling challenging. The process typically involves the pre-treatment of WPCBs, separating the electronic components (ECs) from the boards, and then subjecting both the ECs and waste PCBs to various recycling processes, such as mechanical–physical methods, hydrometallurgy, and pyrometallurgy [13–16].

While there is ample research on the recycling of WPCBs and waste PCBs, less attention has been paid to the specific recycling of waste ECs. Waste PCBs contain hundreds of ECs, such as IC chips, resistors, and capacitors, many of which retain their functionality despite the overall loss of function of the PCB. Even though WPCBs no longer serve their primary function, a significant number of waste ECs are still usable. This is because the average lifespan of ECs is around 20,000 h, which is less than 5% of their intended lifespan of 500,000 h [12]. These waste ECs still hold recycling value, with some chips having a higher value than tons of WPCBs. Moreover, certain waste ECs house a significant portion of the rare metal resources found in WPCBs. For example, tantalum capacitors consist of 30–40 wt.% of tantalum, while multilayer ceramic capacitors are enriched with palladium (Pd), and memory chips contain predominantly gold (Au) [17–19]. However, the composition of waste ECs is more complex than that of waste PCBs, often containing toxic substances, and thereby requiring specialized treatment processes.

To determine whether the concept of in situ utilization is applicable universally or not, this article aims to address these questions through a comprehensive analysis and economic modelling based on real cases. The analysis is based on a thorough review of relevant studies obtained from reputable databases including Web of Science, Scopus, and

Google Scholar. The search for relevant studies was conducted using specific keywords such as e-waste, waste ECs, dismantling, sorting, WPCB, pollutant release, and recycling. After carefully reviewing the articles, we noticed that no significant attention was paid to the economic assessment of pre-treatment methods. Furthermore, we have also taken into consideration recent review articles on the topic of waste ECs recycling. However, we found that only two review articles were available, both of which primarily focused on pre-treatment methods—dismantling techniques and metal separation technologies [20,21]. Unfortunately, other critical aspects of EC recycling, such as pollutant release and treatment, high-value recycling, environmental and economic analyses, and typical pilot recycling processes, have not been thoroughly reviewed in the existing literature. This highlights the need for a comprehensive review that covers these underexplored areas in order to provide a more holistic perspective on waste ECs recycling.

This emphasizes the necessity of conducting a thorough investigation into the pre-treatment of PCBs and conducting economic assessments of this particular aspect. By addressing this relatively unexplored area, we will be able to offer a comprehensive and well-rounded approach to waste ECs recycling. This research will contribute to a better understanding and implementation of efficient recycling methods in the future.

### 1.1. Precious Metals Price Tendency

Over the past three years, the prices of precious metals, such as silver (Ag), gold (Au), and palladium (Pd), have experienced varying trends. From 2019 to early 2020, the price of silver remained relatively stable, with some minor fluctuations (Figure 1). However, in March 2020, as a result of the global pandemic and economic uncertainties, the price of silver experienced a significant decline. Following this drop, silver steadily increased in value throughout the rest of 2020 and into 2021, reaching new highs driven by increased industrial demand, investor interest, and inflation concerns.

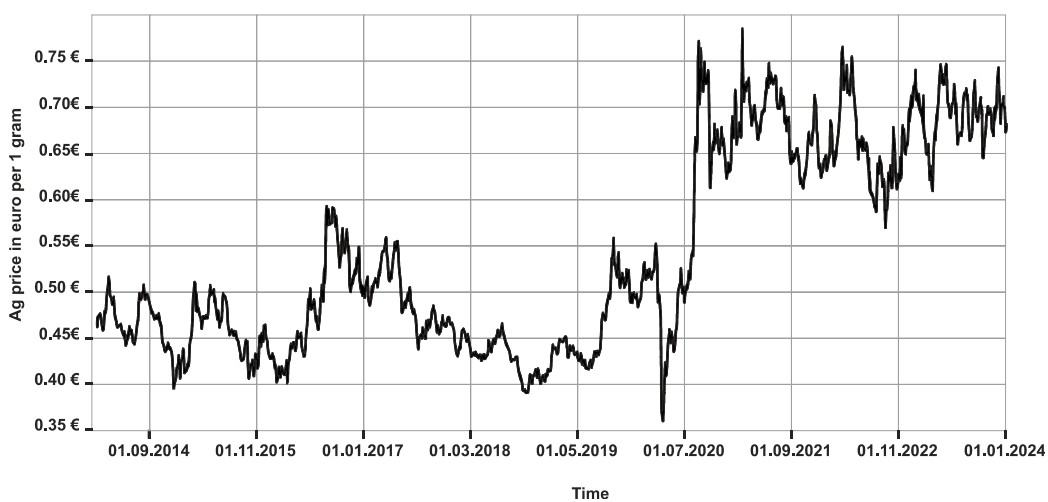

**Figure 1.** Silver price in EUR per gram for last 10 years. Adapted from [22].

Gold, on the other hand, showed a generally upward trend over the past three years (Figure 2). In 2019, gold had a modest price increase, reaching new highs in mid-2020 due to the pandemic-induced economic uncertainties. The metal continued to perform well throughout 2020, driven by low interest rates, a weaker US dollar, and safe-haven demand. However, as the global economy began recovering in 2021, gold experienced some price corrections, but it still maintained strong investor interest as a hedge against inflation and a store of value.

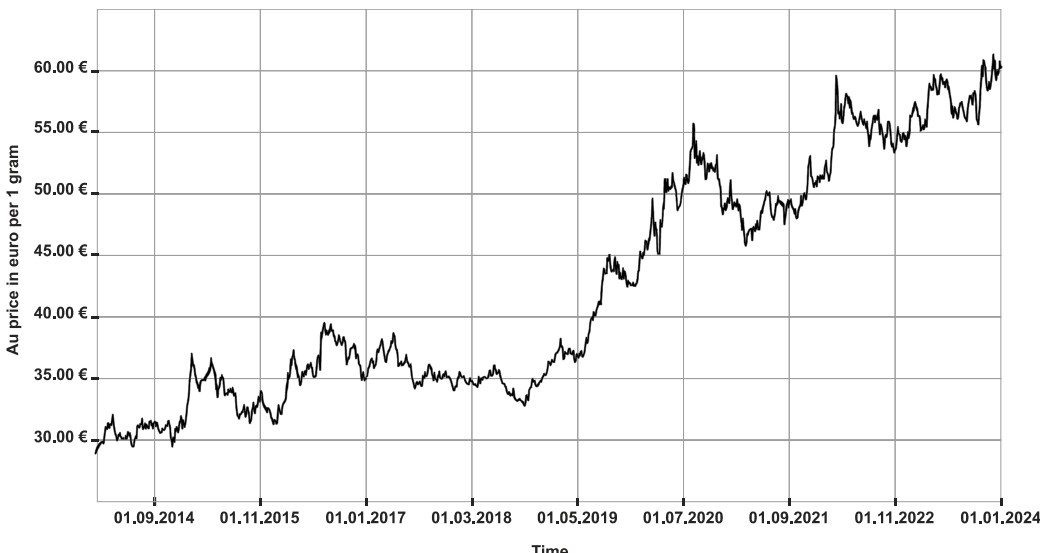

**Figure 2.** Gold price in EUR per gram for last 10 years. Adapted from [23].

Palladium had a remarkable price surge over the past three years. In 2019, the metal experienced a significant increase in value, driven by increasing demand from the automotive industry for catalytic converters (Figure 3). This trend continued into 2020, with palladium reaching all-time highs due to supply deficits and stringent emission regulations. However, in 2021, palladium faced some price corrections as automakers looked for alternatives and supply constraints eased. This impacted prices more significantly in 2023—the price rapidly decreased from EUR 75 to 35 per gram (Figure 3). Nonetheless, palladium remains a valuable and sought-after precious metal.

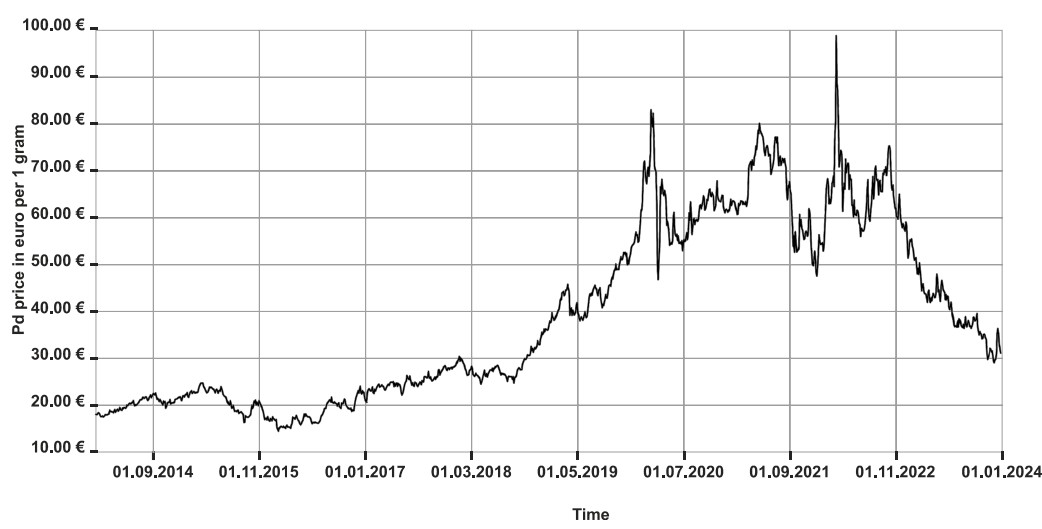

**Figure 3.** Palladium price in EUR per gram for last 10 years. Adapted from [24].

It is important to note that the prices of precious metals are influenced by a variety of factors, including economic conditions, geopolitical events, supply and demand dynamics, and investor sentiment. Therefore, these trends provide a general overview of their price tendencies, but analyzing the long-term tendency in a 10-year period clearly shows a price growth of almost double in comparison with 2013.

*1.2. Electronic Waste Composition and Statistics*

In 2020, the e-waste monitor reported that an estimated 53.6 Mt of WEEE was generated in 2019 in the world, out of which only approximately 9.29 Mt were documented for

recycling. The EU generated approximately 12 Mt of WEEE in the same year, out of which only 5.1 Mt or 42.5% on average was documented for recycling [25].

In 2019, the majority of electronic waste was generated in Asia, amounting to 24.9 million metric tons. On the other hand, Europe had the highest per capita e-waste generation rate, with each person contributing 16.2 kg of e-waste [26]. These figures represent 9.4% and 8.8% of the global total, respectively. Interestingly, Africa, despite ranking last in overall e-waste generation, had the highest documented formal e-waste collection and recycling rate at 42.5% [26]. This indicates that African countries have implemented effective systems for collecting and recycling e-waste, even with lower overall generation levels. In contrast, on all other continents, the amount of e-waste that is officially collected and recycled is significantly lower than the estimated volume of e-waste generated. For example, in Asia, only 11.7% of the e-waste was formally collected and recycled in 2019 [26]. Similarly, the Americas and Oceania had relatively low rates, standing at 0.9%. It is important to understand that e-waste statistics can vary substantially across different regions due to various factors. These factors include income levels, policy frameworks, and the structure of waste management systems, among others.

According to EU E-waste directive (2012/19/EU) [27], e-waste is represented in six categories:

1. Temperature exchange equipment;
2. Screens, monitors, and equipment containing screens having a surface greater than 100 cm$^2$;
3. Lamps;
4. Large equipment;
5. Small equipment (no external dimension greater than 50 cm);
6. Small IT and telecommunication equipment (no external dimension greater than 50 cm).

European statistics of the WEEE, by clases represented on the Figure 4.

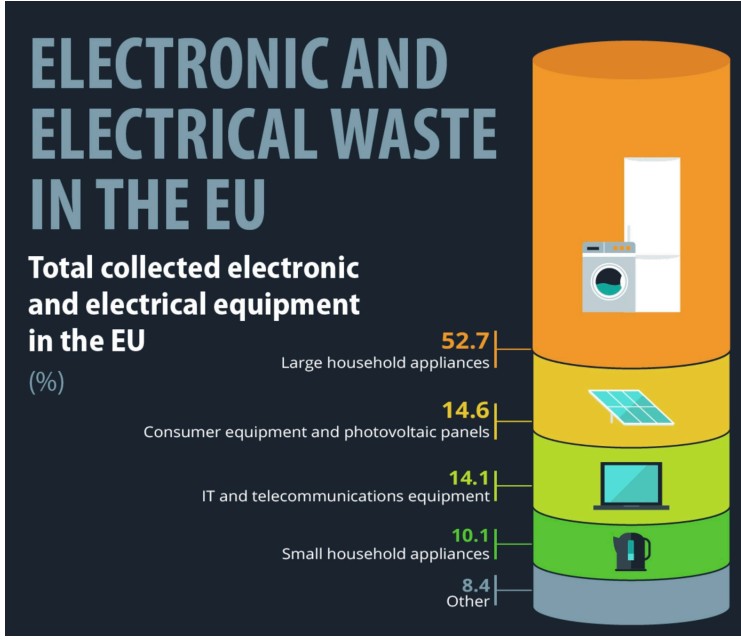

**Figure 4.** The percentage of e-waste per appliance type in the EU by BYCC 4.0 [28].

Among the most priced components in WEEE are the waste PCB (WPCB) rich in rare and precious metals (RPMs), as well as base metals [29]. The WPCB in WEEE constitutes approximately 4–7% of the total WEEE weight, according to Gonzalez et al. [30]. A detailed overview by elements of small IT and telecommunication equipment is given in Table 1.

Higher-grade mines have densities of 8.0 to 10.0 g/t, while lower-grade mines have densities of 1.0 to 4.0 g/t. As follows from Table 1, the Au concentration in electronic waste (type 6) is two-to-three orders of magnitude higher than in ores, reaching up to 1740 g·t$^{-1}$. Moreover, this source of valuable metals is accumulated in industrial zones (cities) and does not need to move to extraction factories or a new building (in case of a rich deposition). The use of waste for the recovery of valuable elements is known as urban mining [31].

**Table 1.** Content of valuable metals in PCBs (g·t$^{-1}$). MP = mobile phone, M = modem, MB = motherboard, CPU = central process unit, MIX = mixed PCBs. - = not analyzed. Adapted from [32].

| Element | MP | M | MB | CPU | MIX | Element | MP |
|---------|-----|-----|-----|-----|-----|---------|-----|
| Cu | 290,120 | 164,950 | 211,380 | 166,333 | 287,300 | Cu | 290,120 |
| Zn | 4680 | 11,820 | 670 | - | 502 | Zn | 4680 |
| Cd | 31 | 360 | 130 | - | 360 | Cd | 31 |
| Ni | 15,740 | 14,140 | 2810 | 78,237 | 6143 | Ni | 15,740 |
| Pb | 14,450 | 29,010 | 18,030 | - | 27,342 | Pb | 14,450 |
| Fe | 31,610 | 57,580 | 1810 | - | 9900 | Fe | 31,610 |
| Cr | 1310 | 250 | 70 | - | 3620 | Cr | 1310 |
| Si | 96,610 | 134,600 | 103,430 | - | 110,000 | Si | 96,610 |
| Al | 19,810 | 36,230 | 18,980 | - | 10,200 | Al | 19,810 |
| Au | 1740 | 21 | 120 | 3270 | 853 | Au | 1740 |
| Ag | 1210 | 1760 | 660 | 1 | 425 | Ag | 1210 |
| Sn | 28,540 | 62,160 | 33,410 | 1 | 55,500 | Sn | 28,540 |
| Sb | - | - | - | - | 1067 | Sb | - |
| Mn | 13 | - | - | - | 33 | Mn | 13 |
| Pd | 125 | - | - | - | 250 | Pd | 125 |
| Pt | 7 | - | - | - | 12 | Pt | 7 |

*1.3. Recovery Technologies*

Recovering precious metals, such as silver (Ag), gold (Au), and palladium (Pd), from PCBs is an essential process in electronic waste recycling. Various technologies are employed to extract these valuable metals from PCBs. One common method is mechanical processing, where PCBs are shredded, crushed, and ground into fine particles. This allows for the separation of different components, including precious metal-bearing parts.

Mechanical processing is often considered one of the stages of pre-processing after preliminary disassembling, dismantling, and further recovery of valuable materials/metals. Mechanical processing includes chopping, shredding, crushing, milling, etc. These are followed by various separation methods based on the difference in properties between metallic and non-metallic components, such as magnetic separation, eddy current separation, separation based on density and electrical conductivity, etc. [33–36]. In addition, the use of selective flotation can be a good alternative to traditional separation methods, especially for the separation of plastic components in electronic waste [37].

Subsequently, hydrometallurgical techniques such as leaching are employed to dissolve the metals from the PCB materials. Acid- or cyanide-based solutions are often used to selectively extract the precious metals. Once the metals are in solution, they can be further purified and recovered through processes such as precipitation, electrolysis, or ion exchange. Advanced methods, such as bioleaching using microorganisms, have also shown potential for precious metal recovery from PCBs. These technologies contribute to the efficient and sustainable utilization of precious metals, reducing the need for primary mining and minimizing the associated environmental impacts. Table 2 summarizes the most widespread pre-treatment methods and their associated advantages and disadvantages.

**Table 2.** Most widespread WPCB/WPCBA pre-treatment methods and their associated advantages and disadvantages.

| WPCB/WPCBA Pre-Treatment Methods | Example(s) | Output(s) | Advantages | Disadvantages |
|---|---|---|---|---|
| Manual dismantling [38–40] | Manual disassembly work (individual or chain) | WPCB/A in their initial state EC | Low investment cost; Utilize simple tools; Job creation for low educated workers; Can be performed selectively/simultaneously; Highest recovery efficiency (well preserved WPCB/A); Small scale; Suitable for developing countries; Selective dismantling. | Lowest copper concentrate quality; Low speed; Health issues; Labor intense; High recovery efficiency; Small scale; Highest OPEX per ton of concentrate; Ergonomically limited. |
| Traditional methods [38–40] | (Incineration) | Dioxins $CO_x$, $NO_x$; $SO_x$ PBDEs HBCDs | Short process time; Simple; No investment required; No equipment is required; No pre-treatment is required; No capacity limitation (hill size fire). | Banned in the EU; High toxicity; High impact on environment and health; Low concentrate quality; Risk of losing control over reaction; Only metals recovered; Low/none energy conversion rate (energy is lost). |
| Physical [38–40] | Milling; Shredding; Electrostatic separation; Air (inertial, centrifugal, gravity); Flotation (water solutions). | Metal dry concentrate Non-metal dry concentrate Dust fraction (K, Br, Cl) Odor (Cl, Br, etc.) Water residues (only in wet separation) | Simplicity (does not require dedicated training); Relatively high production rates; Selective (can target specific group of materials). | Lowest concentrate purity (only manual is lower). |
| Flotation: reverse, alkali [41–44] | Physical separation of non-metal parts by flotation method | Metal wet concentrate (sink part). Non-metal wet concentrate (float part) | Increased recovery efficiency of base metals (Cu, Al, Zn) and Ag [42]. | Methods generate significant amount of waste water; Contaminated by flotation agents (laurylamine, diesel oil, other active components); Not efficient for increasing Au, Pd, and Pt extraction; Float and sink part contaminated by floating media. |
| Bioleaching [38,40,45] | Biomining using microbes: Autotrophs Heterotrophs Mesophilic Thermophilic | Copper foils; Gasses ($CO_2$, $CH_4$, etc.); Glass fibers, ceramics; Liquor. $CO_2$ | Weak organic acids are used; Eco-friendly (green technology). Suitable for both base and precious metals extraction; Low temperature and energy requirement; Clean nonmetal product; Low investment/operating cost; Cost-effective; Selective recovery; Less natural gas and water required. | Difficulty in microorganism isolation; Difficulty in microorganism reproduction/culture. Requires nutrients for microorganisms; Selective to specific metals; Vulnerability to heavy metals (needs selective pre-treatment?); Small scale (scalability constraints); Bacteria toxicity; Low leaching speed; Slow leaching kinetics; Long process time (48–245 h). |



**Table 2.** *Cont.*

| WPCB/WPCBA Pre-Treatment Methods | Example(s) | Output(s) | Advantages | Disadvantages |
|---|---|---|---|---|
| Chemical [38,46] | SCF (super critical fluids), Leaching, Ions exchange, etc. | Copper concentrate; Liquor emissions (e.g., $HNO_3$, $HClO_4$-based) Water solutions (residues). | Highest quality of end-products (metals recovered); Selective in terms of target materials (e.g., dedicated gold recovery); Lower gaseous emissions compared to thermal treatment (in case of SCF could even consume COx for reactions). | Corrosive; Requires reagents and their subsequent recycling; Often requires pre-treatment and concentration for the input (e.g., mechanical or thermal); Often energy consuming (SCF case); Cannot recover non-metals. |
| Chemical (mechanical pre-treatment) [38,45] | Leaching/SCF and physical pre-treatment. | Physical and chemical combined. | Physical and chemical combined. | Physical and chemical combined. |
| Electro-mechanical [38,45,47,48] | (HVF, HVP) | Liquor (waster residues from the peeling of the epoxies, etc.), Copper clad. | Highest ration of powder size/purity among mechanical and combined methods (98% at +3.0 mm size); Low risk of losing precious/noble metals (the Au, Pt, Pd, etc.) coating remains mostly intact. | Relatively high electric energy consumption (5 times average mechanical size reduction); Relatively low process capacity. |
| Thermal [38,39,45] | (Pyrolysis, smelting, microwave, etc.) | Copper matte; Solid by-products (e.g., iron-silica, fly ash, etc.); Gaseous emissions (COx, NOx, SOx, BrO, etc.). | Quality/speed ratio for enrichment is the best among all; Incinerated fraction can be converted to heat/el. energy. | Highest amount of emissions; High CAPEX; High OPEX; Requires dedicated training; Requires more operational permissions; Recovery of plastics is not possible; Fe and Al oxides end up in slags; Lightweight dust fraction containing metals could be burned before reaching metal bath. |
| Thermal-mechanical [38,39,45] | Desoldering (IR, bath, etc.) pre-heating and physical separation | Electrical components (chipsets, resistors, etc.) Substrate plate (copper clad laminate) Solder Emissions (COx, NOx, etc.). | Accurate recovery of ECs Better homogenization of the separation process input -> less materials losses and emissions Allows partial reuse of ECs. | Approximately 20% higher OPEX than fully physical pre-treatment due to thermal depopulation; Lower production speed (desoldering is generally slow ~150 kg/h); Thermal treatment leads to epoxy evaporation and odor generation in higher pace. |

Through the authors' understanding, and by summarizing the literature data and practical experience of the EIT RawMaterials "RENEW: recycling of epoxies from e-waste" project implementation, we proposed an end-of life EEE and WEEE materials flow chart by stages (Figure 5).

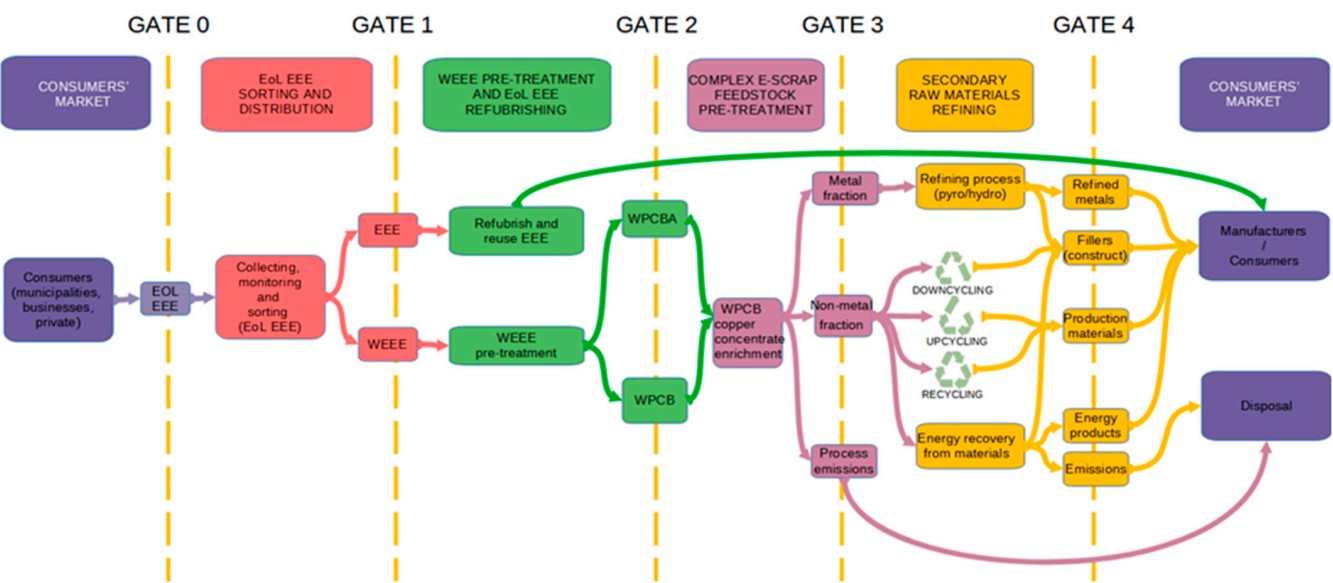

**Figure 5.** End-of life EEE and WEEE materials flow chart by stages. Scheme proposed by authors.

The whole sequence can be segmented into five stages (gates 0–4):

- Gate 4–0: consumer market;
- Gate 0–1: End-of-Life (EoL) consumer goods collection and sorting into EoL Electronic and Electric Equipment (EEE) for reuse and WEEE for recycling (or disposal);
- Gate 1–2: WEEE pre-treatment and disassembly to its basic components, removing hazardous and directing recyclable components to materials (metals, plastics, etc.) recovery.
- Gate 2–3: E-scrap feedstock pre-treatment generally performed either at WEEE pre-treatment facilities or at raw materials recovery facilities (e.g., metallurgy). At this stage the components are being break down to the basic materials and sorted into fractions (concentrates).
- Gate 3–4: Secondary raw materials recovery (re-, up-, downcycling) through metallurgy, plastics remelting, ceramics recycling, etc.

### 1.4. Electronic Waste Availability (Accumulated Quantity) by Markets

Figure 6 shows the market segmentation for WPCB pre-treatment for further metal recovery utilizing pyro- and hydrometallurgical means.

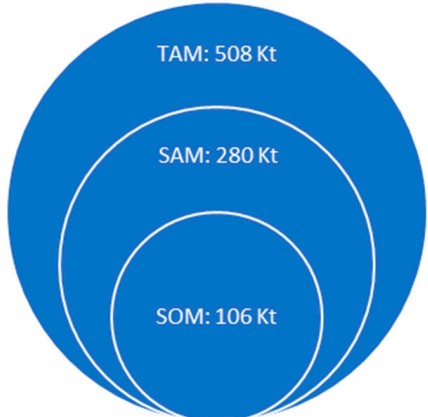

**Figure 6.** Assuming WPCBs form 4–7% (5.5% average) from total WEEE collected for recycling weight. TAM: World WPCBs e-scrap; SAM: EU WPCBs e-scrap; SOM: Baltic sea region (Nordics, Baltics, Germany, Poland) [26].

In this research, we have used a literature review to collect data on existing methods of PCB recycling, identifying the capacity of the market, types of PSBs, and recycling methods. Based on the collected data, we have made our own case study on PCB pre-treatments. This qualitative research method can provide valuable insights and real-world examples. This investigation seeks to bridge the knowledge gap and provide valuable in-sights that will facilitate a more profound understanding of the intricacies involved in efficient recycling methods. Ultimately, the outcomes of this research hold the potential to not only enhance our comprehension but also to drive the practical implementation of sustainable and economically viable recycling solutions in the future.

It is very important to note that in the article, the authors examine metal concentrates that accumulate in fine metal-rich fractions (less than 350 microns), which are obtained by disintegrating a PCB. Attention was paid to the economy of extraction of precious metals using electrochemical methods in combination with disintegration as a pre-treatment. However, it is worth considering that, simultaneously with this process, the accumulation and concentration of metal components occurs in coarse fractions, which were separated and were not further studied here. In addition to precious metals, these large fractions also contain a number of other valuable elements, such as non-ferrous metals (Cu, Ni, Zn, Pb, Sn, etc.), the extraction and return of which to the market will increase the added value of PCB processing and, accordingly, the attractiveness of their processing. The main goal of the paper is to estimate the total energy consumption for the developed technology (milling and leaching) and determine the reasonability of further research and development. The novelty of this research lies in the fact that a technical and economic analysis has been carried out on a newly developed combined technology for processing electronic waste, which includes mechanical processing and electrochemical leaching with the help of active chlorine that is formed in situ. Real (not specially selected or prepared) waste PCBs were used for the economic efficiency evaluation of the process.

## 2. Materials and Methods

In this work, the authors considered only the cost of electricity, labor costs for PCB pre-treatment, and valuable metals leachate preparation. When calculating the full estimate of the cost of such processing, it will also be necessary to add the cost of repairing the rotors (compensation for wear of milling impact bodies/impact elements).

The cost of repairing rotors depends on many factors, such as the finger material (hard alloy, hard-alloy surfacing), metal concentration in the PCBs, impact velocities, etc. This issue will require additional research for the selection of the optimal grinding separation process, which is a topic for a future article.

For the self-cost estimation of the innovative approach, PCB complex pre-treatment, which includes selective disintegration [49] and electrochemical metal leaching—electro-hydro-chlorination (alternating current action in hydrochloric acid electrolytes) [50,51], was used, as it was previously approved and described in detail in previous studies [49–51]. PCBs were subjected to disintegration once as raw materials. Obtained particles smaller than 2.8 mm were subjected to a metal analysis study and designated as X1 (Figure 7). Subsequently, particles larger than 2.8 mm were subjected to repeated milling and designated as X1 (>2.8) + X2 (see Figure 7). Figure 7 represents the PCB milling scheme. Figures 8 and 9 show studied PCB P3 and P4 as supplied and after milling.

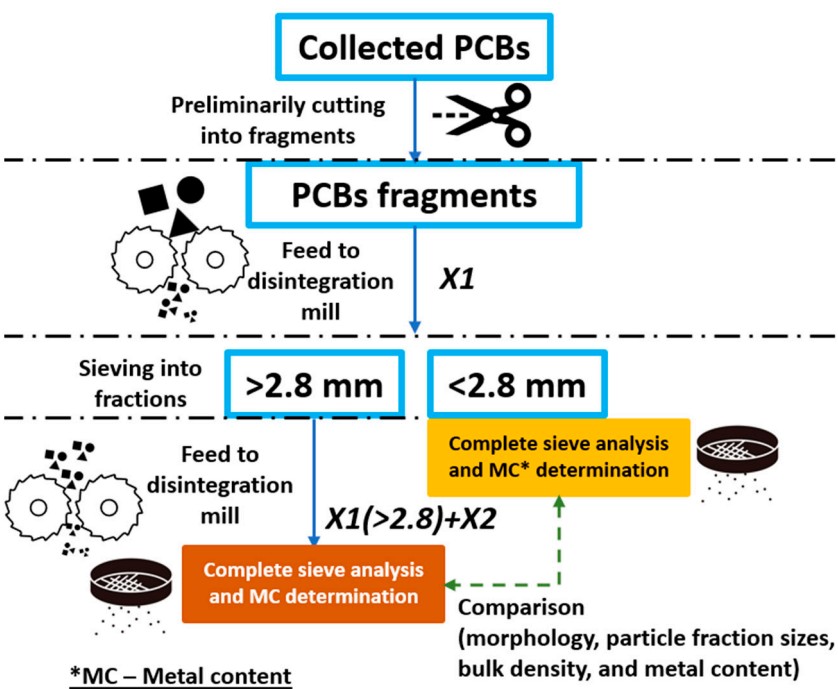

**Figure 7.** The disintegration milling, sieving, and testing scheme of PCB pre-treatment for metal concentrate obtained using one-step (*X1*) and double-step (X1(>2.8) + X2) disintegration milling [49]. Reprinted on CC BY 4.0 open access basis.

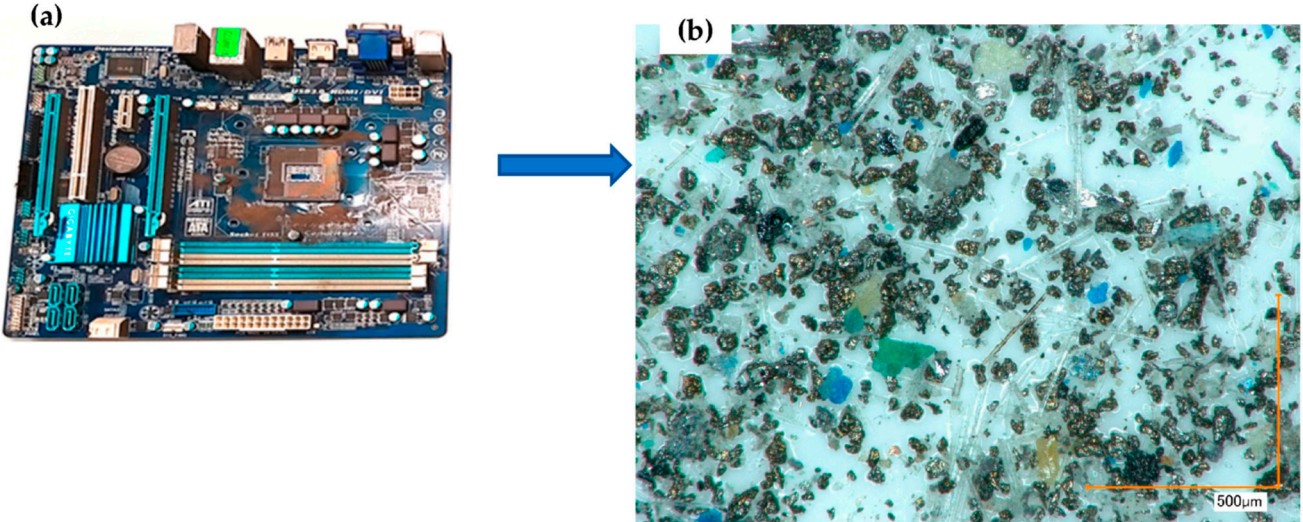

**Figure 8.** Image of studied PCB "P3" (computer motherboard without processor) (**a**) and optical image at magnification ×200 of disintegrator-crushed metal concentrate (fraction < 90 μm) (**b**) [50]. Reprinted on CC BY 4.0 open access basis.

To establish metal content, the chemical treatment (leaching) of three representative samples (0.500 g each) was carried out. The electrochemical hydrochlorination was carried out according to developed technology, where active chlorine is generated in situ during the alternate current electrolysis [49,50]. ICP-OES was used for the quantitative determination of metal content in electrolyte solutions obtained as a result of electrochemical leaching. Based on the results of the analysis, an average concentration of each metal under study was calculated.

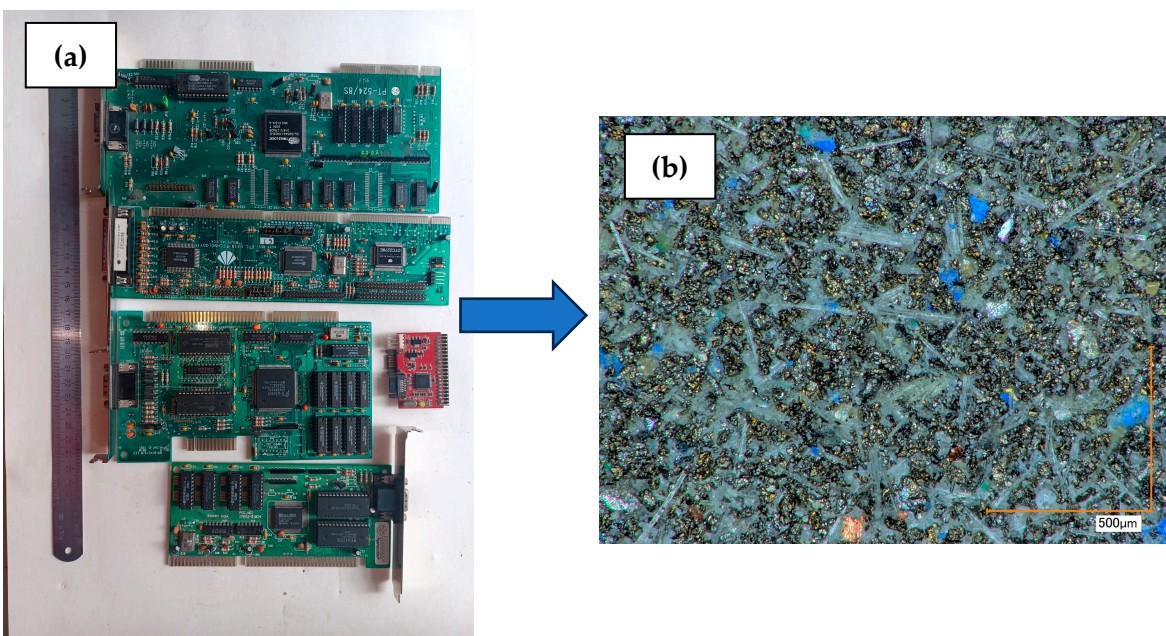

**Figure 9.** Image of studied PCB "P4" (mixed PCB with microchips) (**a**) and optical image at magnification ×200 of disintegrator-crushed metal concentrate (<90 μm) (**b**).

### 3. Results

The valuable metal content strongly depends on the raw PCB's nature (type of electronics and year of its production, among others). To analyze the expected economic effect, the authors analyzed two cases of different PCB types. The first was a computer motherboard and the second was a mixed PCB (Figures 8 and 9).

According to our previous research [49,50] using a pre-treatment of mechanical disintegration of the PCB, and using for the valuable metals extraction fractions with its highest concentrations—> 90, 90–180 and 180–350 μm particles size. The fractions, fractions content to whole PCB (Raw PCBs) and metal content in wt.% are given by fractions and fraction part from initial PCB amount in the Table 1. For the one industrial batch simulation 10,000 kg of raw PCB was taken for estimation. P3-1, P3-2, P4-1 and P4-2 correspond to PCB P3 and P3 for primary and secondary milling respectively. In Table 3 provided Ag, Au, Pd and Cu content in P3-1, P3-2, P4-1 and P4-2 studied materials in wt. % and calculated for test batch of the 10,000 kg, which are used in further calculations.

For the price estimation, 10 metric tons (10,000 kg) of PCB was used as an example. For the pre-treatment of 10 t of PCB, the energy necessary for the first stage of milling (P3-1 and P4-1) was calculate using Formula (1):

$$M \times Emil = Eabs \tag{1}$$

where M is the treated material quantity in tons, Emil is the milling specific energy in kWh/t, and Eabs is the absolute milling energy in kWh. For 10 tons, this results in 160 kWh. Using the current price of EUR 0.16/kWh including VAT at 21% (local energy provider ENEFIT Ltd., Riga, Latvia), this gives (160 kWh × EUR 0.16/kWh) EUR 25.60. For the second stage of milling (P3-2 and P4-2), the 9.2 tons (0.8 t was separated) of material after the first milling calculated using formula (1) (9.2 t × 16 kWh/t) results in 144 kWh. At the current price this gives (144 kWh × EUR 0.16/kWh) EUR 23.04. Additionally, we added to this the conveyor 0.64 kWh/t and screening (1.15 kWh/t) energy consumption (0.64 + 1.15) × (10 t + 9.2 t), which gave 34.36 kWh, or EUR 5.50 Euro.

**Table 3.** The Ag, Au, Pd and Cu content in P3 and P4 concentrate in mass % and absolute amount in kg for the industrial batch simulation of 10,000 kg of raw PCB.

| | | **Raw Material Powder P3-1 (Motherboard, Single Crushing)** | | | | | | | | |
|---|---|---|---|---|---|---|---|---|---|---|
| **Fraction Size, μm** | **Fraction from Raw PCB, %** | **Metal Content in the Fraction, %** | | | | **Metal Content in kg in Test Batch of 10,000 kg** | | | | |
| | | **Ag** | **Au** | **Pd** | **Cu** | **Ag** | **Au** | **Pd** | **Cu** | |
| <90 | 7.100 | 0.086 | 0.0016 | 0.0002 | 1.796 | 0.6106 | 0.01136 | 0.0014 | 12.752 | |
| 90–180 | 0.900 | 0.0824 | 0.0002 | 0.0004 | 8.382 | 0.0742 | 0.00018 | 0.0004 | 7.544 | |
| 180–350 | 1.400 | 0.0682 | 0.0002 | 0.001 | 16.836 | 0.0955 | 0.00028 | 0.0014 | 23.570 | |
| | | | | | | 0.780 | 0.012 | 0.003 | 43.866 | Total P3-2, kg |
| | | **Raw material powder P3-2 (Motherboard, double crushing)** | | | | | | | | |
| **Fraction size, μm** | **Fraction from raw PCB, %** | **Metal content in the fraction, %** | | | | **Metal content in kg in test batch of 10,000 kg** | | | | |
| | | **Ag** | **Au** | **Pd** | **Cu** | **Ag** | **Au** | **Pd** | **Cu** | |
| <90 | 0.795 | 0.076 | 0.003 | 0 | 0.407 | 0.060 | 0.0024 | 0 | 0.324 | |
| 90–180 | 0.409 | 0.116 | 0.003 | 0.001 | 5.476 | 0.047 | 0.0012 | 0.00041 | 2.240 | |
| 180–350 | 0.841 | 0.127 | 0.001 | 0.002 | 14.837 | 0.107 | 0.0008 | 0.00168 | 12.478 | |
| | | | | | | 0.215 | 0.004 | 0.002 | 15.041 | Total P3-2, kg |
| | | | | | | **0.995** | **0.016** | **0.005** | **58.907** | **Sum of P3-1 and P3-2, kg** |
| | | **Raw material powder P4-1 (Mixed PCBs, single crushing)** | | | | | | | | |
| **Fraction size, μm** | **Fraction from raw PCB, %** | **Metal content in the fraction, %** | | | | **Metal content in kg in test batch of 10,000 kg** | | | | |
| | | **Ag** | **Au** | **Pd** | **Cu** | **Ag** | **Au** | **Pd** | **Cu** | |
| <90 | 3.600 | 0.97 | 0.1 | 0.17 | 1.75 | 3.492 | 0.360 | 0.612 | 6.300 | |
| 90–180 | 2.100 | 0.56 | 0.02 | 0.1 | 2.69 | 1.176 | 0.042 | 0.210 | 5.649 | |
| 180–350 | 3.200 | 0.51 | 0.01 | 0.04 | 6.1 | 1.632 | 0.032 | 0.128 | 19.52 | |
| | | | | | | 6.300 | 0.434 | 0.950 | 31.469 | Total P4-1, kg |
| | | **Raw material powder P4-1 (Mixed PCBs, double crushing)** | | | | | | | | |
| **Fraction size, μm** | **Fraction from raw PCB, %** | **Metal content in the fraction, %** | | | | **Metal content in kg in test batch of 10,000 kg** | | | | |
| | | **Ag** | **Au** | **Pd** | **Cu** | **Ag** | **Au** | **Pd** | **Cu** | |
| <90 | 3.145 | 0.46 | 0.03 | 0.02 | 2.06 | 1.447 | 0.094 | 0.063 | 6.479 | |
| 90–180 | 1.501 | 0.45 | 0.01 | 0.01 | 4.29 | 0.675 | 0.015 | 0.015 | 6.438 | |
| 180–350 | 2.900 | 0.09 | 0 | 0 | 6.84 | 0.261 | 0 | 0 | 19.839 | |
| | | | | | | 2.383 | 0.109 | 0.078 | 32.757 | Total P4-2, kg |
| | | | | | | **8.683** | **0.543** | **1.028** | **64.26** | **Sum of P4-1 and P4-2, kg** |

According to the obtained experimental data and extrapolating it to 10 tons of raw PCB, after milling and separation for further extraction, a total of 1144.0 kg from the P3 sample and 1644.0 kg from the P4 sample of concentrated milled PCB will be prepared.

The next stage involved the electro-hydrochlorination of milled PCB in an electrochemical for metal transfer into soluble salts (leaching) at two sets of conditions: (a) time 2 h,

I = 120 A, U = 8 V, m PCBs = 0.250 kg; and (b) time 1 h, I = 135 A, U = 12 V, m PCBs = 0.250 kg. This was calculated according to Formula (2):

$$(I \times U \times T)/M = Eh\text{-}Cl, \tag{2}$$

where I is the current strength in A, U is the voltage in V, T is the process time in hours, M is the treated material mass in kg, and Eh-Cl is the specific energy spent for the electro-hydrochlorination process in Wh/kg. For the first case, the spent energy was 7.68 kWh/kg and for the second it was 6.48 kWh/kg. The second case was more beneficial due to its shorter processing time by half, but it led to extensive heat emitting of the electrolyte and needed intensive cooling. Therefore, a further efficiency estimation was taken for the first case—7.68 kWh/kg. This was calculated according to Formula (3):

$$M \times Eh\text{-}Cl = Eabs, \tag{3}$$

where M is the treated material quantity in kg; Ee-Cl is the specific energy spent for the electro-hydrochlorination process in kWh/t, and Eabs is the absolute energy in kWh spent for the process. For the electro-chemical treatment process the consumption was as follows: for P3-1 + P3-2 it was 7.68 kWh/kg × 1140 kg = 8755 kWh or EUR 1400; for P4-1 + P4-2 it was 7.68 kWh/kg × 1644 kg = 12,626 kWh or EUR 2020.

However, in a further scale-up of the process, the tendency of the decreasing of the electricity consumption per 1 kg was clearly seen: for the lab scale (time 2 h, m PCBs = 3 g, I = 8 A, U = 9.4 V) this was noted as 50 kWh/kg, but for 250 g of PCB it was an order of magnitude lower. Further electrolytic cell development and optimizations (electrode geometry optimization and working volume geometry optimization) will lead to an electric current consumption and overall price cost decrease.

The total self-cost of electricity for the PCB transferring into leachate would be EUR 1445.14 for P3 and EUR 2074.14 for P4.

Additionally, human resources of two men-working days for mechanical treatment (EUR 200) and four men-working days for the electro-hydrochlorination process (EUR 400–600) need to be considered.

Table 4 provides an estimation of the theoretically possible metals market value in pure form obtained using the developed technology.

**Table 4.** Possible metals market value in pure form obtained from P3 and P4 raw materials.

| Metals Amount, kg in 10,000 kg of Raw PCB | | | Market Price in EUR/kg | Market Value in EUR | |
|---|---|---|---|---|---|
| Metal | Source P3 | Source P4 | | Source P3 | Source P4 |
| Ag | 0.995 | 8.683 | 660 | 656.7 | 5730.78 |
| Au | 0.016 | 0.543 | 56,000 | 896 | 30,408 |
| Pd | 0.005 | 1.028 | 35,000 | 175 | 35,980 |
| Cu | 58.907 | 64.226 | 8 | 471.256 | 513.808 |
| | | *Total* | | *2198.956* | *72,632.588* |

Challenges and Obstacles:

As can be seen from Table 4, in the case of 100% (in real cases it is 85–95%) of metal extraction, it is possible to recover valuable metals worth EUR 2199 in the case of P3 and for EUR 72,632 in the case of the P4 PCB. Taking into account that potential income, the preparation of leachate concentrate self-cost is only slightly covered by the potential metal selling price: EUR 1645 vs. EUR 2199. In this case, it would have a negative economic effect on the recycling company. In contrast, the PCB P4 example has a sales price that is 27 times greater than the leachate preparation self-cost: EUR 2674 vs. EUR 72,632.

As can be seen in Table 3, the amount of Cu in the concentrate is significant, both as a wt.% and an absolute amount—471.26 and 513.81 kg for P3 and P4, respectively. However, analyzing the possible income (Table 4) from Cu sales, it is not a significant portion (EUR

513.81 or 0.7%) of the potential income (EUR 72,632.59) in case P4, which is commercially reasonable. In the case of P3, it is 21%, but as was discussed above, the use of the P3-type raw material is commercially unfavorable.

Taking into account that this research and technology validation finished at the metal-rich leachate obtained without further separation and metal purification, the final product was the metal-rich leachate, which could not be sold at the pure metal price. The general calculation of the return on investment of the studied technology was carried out.

Our main assumptions included as a base scenario that precious metals were sold as a concentrate at a price that was 50% of their market value in pure form. Table 5 provides an estimation of the theoretically possible key parameter. For the calculation we used a raw material similar to P4 (Table 4).

**Table 5.** KPIs of production theoretical model.

| Parameter | Value |
| --- | --- |
| WPCB processed | 2400 mt/year |
| Extraction efficiency | 14% |
| CAPEX | 6 mil |
| OPEX | 5.2 mil EUR/year |
| Average total revenue | 8.3 mil EUR/year |
| Unit revenue | 3456 EUR/mt |
| Annual profit | 3.2 mil EUR |
| Payback period | 2 years |

The main assumption was that precious metals as a concentrate are worth 50% of their market value in pure form. Taking into account that the price of the extracted metals can fluctuate, we have provided a graph on possible profit/loss scenarios if the extracted material's price drops from 50% to 20% of the market price. Increasing the selling price by +20% from the base scenario and reaching 70% of the pure metal price significantly increased the profit (Figure 10) and could shrink the payback period.

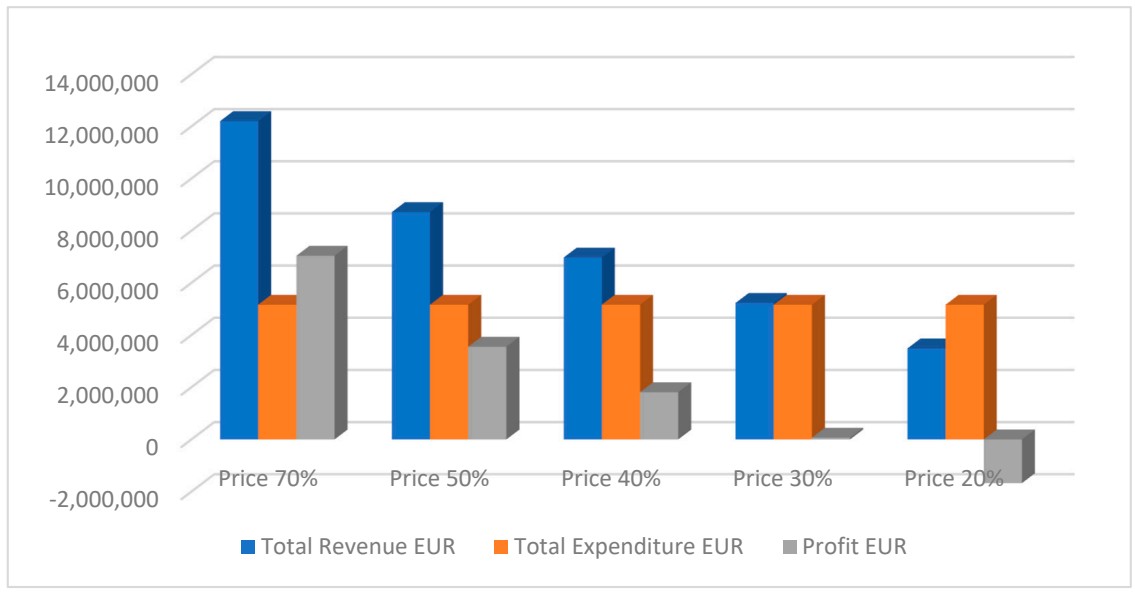

**Figure 10.** Chart of the possible profit/loss depending on extracted material price from market.

As can be seen, in the case of the price drop until 30% the profit drops to zero, and at any lower than 30% it will be negative.

Market Estimation and Future Outlook:

E-waste is a rapidly growing waste stream due to the increasing use of electronic devices. The continuous expansion of technology in various sectors, including consumer electronics, healthcare, and transportation, is expected to contribute to a substantial increase in e-waste. As the global demand for precious metals such as gold, silver, and palladium remains high, concerns about resource scarcity are growing. E-waste recycling plays a crucial role in the circular economy, offering a sustainable and environmentally responsible means to recover and reuse these valuable metals.

Our research has unveiled substantial potential in the field of PCB pre-treatment methods, presenting an opportunity for the reclamation of precious materials. With the capacity to process a minimum of 30% of the total PCBs, equivalent to approximately 75,000 metric tons annually, the potential recovery of precious metals holds an estimated value of EUR 545 million.

In total, e-waste recycling has the potential to reduce the pressure on primary mining operations for precious metals. A steady supply of recycled precious metals can help stabilize the market, decrease price volatility, and mitigate the geopolitical risks associated with mining operations.

## 4. Conclusions

In conclusion, this research underscores the crucial importance of delving into the pre-treatment of PCBs and conducting in-depth economic assessments within this specific domain. By dedicating attention to this relatively unexplored facet of electronic waste recycling, we are taking a significant step towards offering a more holistic and inclusive approach to managing waste electronic components.

As we unlock the untapped potential in this field, we move closer to realizing a more sustainable and responsible management of electronic waste on a global scale.

The results from research further suggested that the roadmap of physical pre-treatment and electro-chemical reclamation could have potential financial benefits and, hence, provide a business opportunity.

As shown by the technical economic estimation, developed technology which combines a disintegration pre-treatment in combination with electro-chemical hydrochlorination would be economically justified with a selling price of metal-rich concentrate at 30% of its market price in pure form under conditions of a 2 year return on investment. A mandatory condition for the process's positive outcome is a high enough valuable metal content.

With a single separation plant capable of producing approximately 5704 t/year of metal concentrates containing precious metals, EUR 41.5 million of precious metals could be reclaimed.

The optimization of hydrochlorination allows the operational costs to be cut in the long-term by 20–30%.

The use of physical separation as a pre-treatment would enable an increase in the electro-chemical reclamation process capacity of approximately 40–60%.

Meanwhile, physical separation ensures the hazardous substances are contained accordingly and further utilized without taking part in the reclamation process, as provided in RoHS Directive 2011/65/EU on the regulation of the hazardous substances in electronic and electric wastes.

**Author Contributions:** Conceptualization E.B., K.M. and A.S.; methodology D.G., K.M. and A.S.; validation V.P., K.M. and J.B.; formal analysis A.K. and A.Z.; investigation E.B., V.S. and D.G.; resources, E.B., V.P. and V.S.; data curation J.B., V.A. and A.Z.; writing—original draft preparation E.B., A.S., K.M., J.B. and D.G.; writing—review and editing V.S., J.B. and A.K.; visualization A.K., V.A. and J.B.; supervision, E.B. and A.S.; project administration E.B. and V.P.; funding acquisition, E.B. and D.G. All authors have read and agreed to the published version of the manuscript.

**Funding:** This research was supported by ERDF project Nr. 1.1.1.1/20/A/139 "Development of sustainable recycling technology of electronic scrap for precious and non-ferrous metals extraction".

The article was published with the financial support from the Riga Technical University Research Support Fund. This article is supported by EIT RawMaterials, which is supported by the European Institute of Innovation of Technology (EIT), a body of the European Union, and the Environmental Investment Centre Foundation co-funding grant KIK22014. ISSP UL as the Center of Excellence was supported through the Framework Program for European universities, Union Horizon 2020, H2020-WIDESPREAD-01-2016-2017-TeamingPhase2 under Grant Agreement No. 739508, CAMART2 project.

**Data Availability Statement:** All data presented in the paper.

**Conflicts of Interest:** The authors declare no conflict of interest.

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
