# Peer review of "Economic Aspects of Mechanical Pre-Treatment’s Role in Precious Metals Recovery from Electronic Waste"

_metals, doi:10.3390/met14010095_

Round 1

Reviewer 1 Report

Comments and Suggestions for Authors

The introduction should be concise, highlighting the focus of this study.

Comments on the Quality of English Language

The language overall is ok

Author Response

Thank you very much for your efforts.

Reviewer 2 Report

Comments and Suggestions for Authors

In this paper, the crucial importance of delving into the pre-treatment of PCB and conducting in-depth economic assessments within this specific domain were studied. It is an interesting content, but arranged structure needs to be further improved. Therefore, it needs minor revision before it is published in this journal. Some issues should be carefully addressed.

Comments and suggestions:

1. Some data about prediction analysis was disclosed, hence the significance, necessity and relevant works should be added in Abstract.

2. The English of the manuscript should be carefully polished, and some expression should be improved.

3. The authors should explain why electronic waste was chosen for recovering precious metals.

4. Flotation is a useful method to recover precious metals from electronic waste before leaching, so it should be described in the “Introduction”, and several relevant references may be added to support this point, such as Int. J. Min. Sci. Technol. 33 (2023) 519-527 ; Int. J. Miner. Metall. Mater. 30 (2023) 1297-1309.

5. It is suggested to compare the results of the present research with some similar studies which is done before.

6. The conclusions should directly present the quantified research achievements, such as important models, data and indicators, etc .., which must be revised.

Comments on the Quality of English Language

Moderate editing of English language is required.

Reviewer 3 Report

Comments and Suggestions for Authors

The article concerns the recycling of printed circuit boards (PCBs), which constitute 3 to 5% of all electronic waste. The topic is very current and current and fits into the current trend of the circular economy. The work focused on the analysis of the chemical composition of used printed circuit boards, which includes the most valuable metals, i.e. Ag, Au, Pd, Cu, Sn, Pb, Cd, Cr, Zn, Ni and Mn. The article presents an economic analysis of preparing a concentrate containing valuable metals (Ag, Au, Pd) from raw PCB and assesses the impact of the PCB pre-treatment method before extracting valuable metals on the change in extraction costs. The authors focused on the analysis of the disintegration method of 10,000 kg of PCBs, which consisted in high-energy impact of particles of the crushed material, thus causing the separation of metallic PCB components. On this basis, own costs and potential profit in the case of full extraction of valuable metals (Ag, Au, Pd) plus Cu were estimated. It was shown that from 10,000 kg of tested PCBs it is possible to obtain 1,144 and 1,644 kg of a concentrate rich in metals, which should then be subjected to electrochlorination in order to leach metals.

The article is written correctly and has a logical layout adn properly and interesting conclusion.

Author Response

Thank you very much for your efforts.

Reviewer 4 Report

Comments and Suggestions for Authors

Manuscript reviews the situation of metal recovery from PCB. Introduction is deeply written the motivation of metal recovery from PCB, but experimental and R&D are unfortunately poor.

1) based on the wave of the world situation, what do you want to do? please mention more clearly. Please find the problem at present, and then mention concretely to overcome them.

2) how to determine the conc. of metals? 

3) in R&D, it is like a report. please write more clearly the scheme of calculation using mathematical equation. You want to optimize the cost, mention the subjective equation.

Round 2

Reviewer 4 Report

Comments and Suggestions for Authors

Manuscript is revised against the comments of the reviewer.